# A Novel Clock Parameterization and Its Implications for Precise Point Positioning and Ionosphere Estimation

**DOI:** 10.3390/s22093117

**Published:** 2022-04-19

**Authors:** Maxim Keshin, Yuki Sato, Kenji Nakakuki, Rui Hirokawa

**Affiliations:** 1Mitsubishi Electric Europe, 40882 Ratingen, Germany; kenji.nakakuki@meg.mee.com; 2Advanced Technology R&D Ctr., Mitsubishi Electric Corporation, Amagasaki 661-8661, Japan; sato.yuki@dn.mitsubishielectric.co.jp; 3Kamakura Works, Mitsubishi Electric Corporation, Kamakura 247-8520, Japan; hirokawa.rui@dx.mitsubishielectric.co.jp

**Keywords:** clock parameterization, precise point positioning, PPP-RTK model, ionosphere estimation

## Abstract

By convention, IGS precise clock products are computed using the ionosphere-free linear combination. Due to the broad use of IGS products, this convention is exploited in PPP-RTK models not using such a linear combination. So, in different carrier phase combinations, the code hardware biases are contained in different combinations, thus making the problem of separating biases from integer ambiguities more complicated. In this paper, we proposed a novel clock parameterization which allows facilitating this problem. Based on the proposed parameterization, we derived a dual-frequency PPP-RTK model for the undifferenced measurements and assessed this model for the static positioning case in terms of positioning accuracy, convergence, and ambiguity resolution performance. The results showed that a cm-level accuracy level is achievable with the derived models with nearly instant convergence and almost 100% successfully resolved ambiguities. We demonstrated the use of this parameterization for slant ionosphere estimation. We derived the analog of the equation linking the wide-lane, geometry-free, and ionosphere-free biases from the Fast-PPP system and used it to retrieve slant ionosphere information. Our TEC estimates showed some evidence of capability to reach an agreement of 1–2 TECU and the standard deviation of 3–4 TECU with GIM TEC values.

## 1. Introduction

As very well known, IGS precise clock products refer to the ionospheric-free L3 linear combination [1,2]. Mathematically, it is represented as a clock parameterization, which can be written as follows [3]
(1)$$c\delta t=\mathrm{cdt}+\frac{f_{1}^{2}}{f_{1}^{2}-{\;f}_{2}^{2}}K_{1}-\frac{f_{2}^{2}}{f_{1}^{2}-{\;f}_{2}^{2}}{\;K}_{2}\;$$
where dt and δt are the original and newly defined satellite and receiver combined clock error, respectively; $K_{1}{,K}_{2}$ are the code hardware delays at frequencies $f_{1}$ and $f_{2}$. This parameterization was introduced, since, in this case, satellite-dependent hardware code delays are absorbed in satellite clock products, so it is not necessary to apply the inter-frequency bias between the code hardware delays at frequencies $f_{1}$ and $f_{2}$, also known as the Differential Code Bias (DCB), when IGS clock products are used in the standard ionosphere-free PPP approach [4]. This conventional clock parameterization was actual and advantageous in the past, when the ionosphere-free PPP model was the main one for high-precision positioning. Nonetheless, numerous recent multi-frequency, multi-GNSS PPP-RTK models still make use of it, mainly due to the broad use of IGS products, although those models do not rely upon the ionosphere-free linear combination any longer [5,6,7,8,9,10,11]. Let us briefly consider what effect the employment of this clock parameterization has.

Table 1 gives a summary of the carrier phase biases associated with PPP models that use clock parameterization (1). As is seen, the dual-frequency carrier phase combinations have different frequency-dependent code ($K_{1}{,K}_{2}$) and carrier ($k_{1}{,k}_{2}$) hardware delay-induced biases $b_{*}$, of which only the wide-lane bias can be quickly estimated. This makes the problem of ambiguity resolution a difficult task, requiring coping with the phase biases $B_{*}$, which comprise an integer part $\lambda_{*}N_{*}$, corresponding to an unknown number of cycles of wavelength $\lambda_{*}$, and a fractional part corresponding to the biases $b_{*}$. For instance, in the framework of the Fast-PPP approach, the following expression is applied [3,12].
(2)$$\frac{1}{\alpha_{w}}\left(B_{W}-{\;B}_{C}\right){=B}_{I}{=\Phi }_{I}-\;I\;-{\;K}_{21}\;$$
where $I\;=\;\left(\alpha_{2\;}-{\;\alpha }_{1}\right)\cdot \mathrm{STEC}$ with $\alpha_{i}\;=\;\frac{{40.3\cdot 10}^{16}}{f_{i}^{2}}$, $\Phi_{I}$ is the ionosphere-free carrier phase, the phase bias $B_{*}$ can be represented as $B_{*}=b_{*}+\lambda_{*}N_{*}$, $\alpha_{w}\;=\;\frac{f_{1}f_{2}}{f_{1}^{2}\;-{\;f}_{2}^{2}}$ and $K_{21}{\;=\;K}_{2}\;-{\;K}_{1}$ is the inter-frequency bias between the code hardware delays at frequencies $f_{1}$ and $f_{2}$, or DCB. If, on one hand, the ionosphere slant delay I is available and the wide-lane bias $B_{W}$ is known, the ionosphere-free bias $B_{C}$, can be computed and used by a navigation filter. Since that estimate is more accurate than the one obtained from processing the ionosphere-free measurements, it helps to accelerate filter convergence in the classical PPP model. If, on the other hand, the biases $B_{W}$ and $B_{C}$ are available, the geometry-free bias $B_{I}$ can be computed, and so the sum ${I+K}_{21}$. What remains to be done is to de-couple the DCB $K_{21}$ from the ionospheric delay term I to receive unbiased and accurate ionosphere information. The last de-coupling step can be considered a drawback of the approach discussed, and this cannot be changed so far as the clock parameterization (1) is employed. Furthermore, the biases in different carrier phase combinations ($b_{W}{,\;b}_{I}{,\;b}_{C})$ are, in fact, considered as fully independent quantities, which is not the case. The logical step that stems from these conclusions is to consider other clock definitions to overcome this difficulty and to develop PPP/PPP-RTK models with some pre-defined desirable properties. Below, we demonstrate how this can be achieved. We are not going to provide detailed and comprehensive consideration. Instead, we give only a brief overview aimed at demonstrating the clock re-parameterization idea and outline what advantages and disadvantages of ionosphere estimation and user positioning this idea can deliver.

## 2. Novel Clock Parameterization

We begin with the clock definition written in the generalized form
(3)$${c\delta t=\mathrm{cdt}+\beta }_{K1}K_{1}{+\beta }_{K2}K_{2}{+\beta }_{k1}k_{1}{+\beta }_{k2}k_{2}$$
where $\beta_{K1}{,\;\beta }_{K2}{,\;\beta }_{k1}{,\beta }_{k2}$ are arbitrary numerical coefficients. The concrete form of the coefficient will be dependent on additional conditions and properties imposed. In order to formulate the conditions, let us write down expressions for the ionosphere-free and the Melbourne-Wübenna (M-W) combinations using the generalized clock parameterization (3). The ionosphere-free carrier phase and code combination read
(4)$$\begin{array}{l} \Phi_{C}{\;=\;u}_{r\;}-{\;\beta }_{K1}K_{1}\;-{\;\beta }_{K2}K_{2}+\frac{f_{1}^{2}\left(1\;-{\;\beta }_{k1}\right){\;+\;f}_{2}^{2}\beta_{k1}}{f_{1}^{2}\;-{\;f}_{2}^{2}}k_{1}\\ -\frac{f_{1}^{2}\beta_{k2}{+f}_{2}^{2}\left(1\;-{\;\beta }_{k2}\right)}{f_{1}^{2}-f_{2}^{2}}k_{2}{+\lambda }_{N}\left(N_{1}+\frac{\lambda_{w}}{\lambda_{2}}N_{w}\right){+\varepsilon }_{\Phi_{C}}{=u}_{r}{+b}_{C\;}{+\lambda }_{N}\left(N_{1\;}+\frac{\lambda_{w}}{\lambda_{2}}N_{w}\right){+\varepsilon }_{\Phi_{C}} \end{array}$$
(5)$$R_{C}{=u}_{r}+\frac{f_{1}^{2}\left(1\;-{\;\beta }_{K1}\right){+f}_{2}^{2}\beta_{K1}}{f_{1}^{2}-{\;f}_{2}^{2}}K_{1}-\frac{f_{1}^{2}\beta_{K2}{+f}_{2}^{2}\left(1\;-{\;\beta }_{K2}\right)}{f_{1}^{2}-{\;f}_{2}^{2}}K_{2}-{\;\beta }_{k1}k_{1\;}-{\;\beta }_{k2}k_{2}{+\varepsilon }_{R_{C}}$$
where $\lambda_{w}\;$and $\lambda_{N}$ represent the wavelength of the wide-lane and narrow-lane combinations, respectively, $N_{i}$ is the integer ambiguity at frequency $f_{i},$ $N_{w}$ is the wide-lane ambiguity, and $u_{r}{\;=\;\rho }_{r}+\;\left({c\delta t}_{r}\;-{\;c\delta t}^{s}\right){\;+\;T}_{r}$ comprises the receiver and satellite clock errors and user position-dependent parameters: the geometric distance $\rho_{r}$ and the troposphere slant delay $T_{r}$. Similarly, the wide-lane carrier phases and narrow-lane code measurements with the general clock parametrization (3) can be written as follows
(6)$$\Phi_{w}{=u}_{r}{+\alpha }_{w}\cdot I\;-{\;\beta }_{K1}K_{1}-{\;\beta }_{K2}K_{2}+\frac{f_{1}\left(1\;-{\;\beta }_{k1}\right){+f}_{2}\beta_{k1}}{f_{1}-{\;f}_{2}}k_{1}-\frac{f_{1}\beta_{k2}{+f}_{2}\left(1\;-{\;\beta }_{k2}\right)}{f_{1}-{\;f}_{2}}k_{2}{+\lambda }_{w}N_{w}{+\varepsilon }_{w}$$
(7)$$R_{n}{=u}_{r}{+\alpha }_{w}\cdot I+\frac{f_{1}\left(1\;-{\;\beta }_{K1}\right)-{\;f}_{2}\beta_{K1}}{f_{1}{+f}_{2}}K_{1}-\frac{f_{1}\beta_{K2}-{\;f}_{2}\left(1\;-{\;\beta }_{K2}\right)}{f_{1}{+f}_{2}}K_{2}-{\;\beta }_{k1}k_{1}-{\;\beta }_{k2}k_{2}{+\varepsilon }_{n}$$

Using Equations (6) and (7) we can write an expression for the M-W combination with the clock parametrization in the generalized form
(8)$$\begin{array}{c} \Phi_{\mathrm{MW}}=-\left\{\beta_{K1}+\frac{f_{1}\left(1-\beta_{K1}\right)-f_{2}\beta_{K1}}{f_{1}{+f}_{2}}\right\}K_{1}-\left\{\beta_{K2}-\frac{f_{1}\beta_{K2}-f_{2}\left(1-\beta_{K2}\right)}{f_{1}{+f}_{2}}\right\}K_{2}+\left\{\frac{f_{1}\left(1-\beta_{k1}\right){+f}_{2}\beta_{k1}}{f_{1}-f_{2}}{+\beta }_{k1}\right\}k_{1}-\\ \left\{\frac{f_{1}\beta_{k2}{+f}_{2}\left(1-\beta_{k2}\right)}{f_{1}-f_{2}}-\beta_{k2}\right\}k_{2}{+\lambda }_{w}N_{w}{+\varepsilon }_{\mathrm{MW}}=\\ -\frac{f_{1}}{f_{1}{+f}_{2}}K_{1}-\frac{f_{2}}{f_{1}{+f}_{2}}K_{2}+\frac{f_{1}}{f_{1}-f_{2}}k_{1}-\frac{f_{2}}{f_{1}-f_{2}}k_{2}{+\lambda }_{w}N_{w}{+\varepsilon }_{\mathrm{MW}}{=b}_{\mathrm{MW}}{+\lambda }_{w}N_{w}{+\varepsilon }_{\mathrm{MW}}. \end{array}$$

Let us now compare expressions for the biases $b_{C}$ and $b_{\mathrm{MW}}$ in Equations (4) and (8)
(9)$$b_{\mathrm{MW}}=-\frac{f_{1}}{f_{1}{+f}_{2}}K_{1}-\frac{f_{2}}{f_{1}{+f}_{2}}K_{2}+\frac{f_{1}}{f_{1}-f_{2}}k_{1}-\frac{f_{2}}{f_{1}-f_{2}}k_{2}$$
(10)$$b_{C}=-\beta_{K1}K_{1}-\beta_{K2}K_{2}+\frac{f_{1}^{2}\left(1-\beta_{k1}\right){+f}_{2}^{2}\beta_{k1}}{f_{1}^{2}-f_{2}^{2}}k_{1}-\frac{f_{1}^{2}\beta_{k2}{+f}_{2}^{2}\left(1-\beta_{k2}\right)}{f_{1}^{2}-f_{2}^{2}}k_{2}.$$

We now choose the coefficients $\beta_{K1}{,\;\beta }_{K2}{,\;\beta }_{k1}{,\beta }_{k2}$, such that $b_{\mathrm{MW}}$ and $b_{C}$ become equal. This is the main condition we impose to derive the coefficients. It is obvious that, if the coefficients of $K_{1}{,\;K}_{2},k_{1}{,k}_{2}$ in Equations (9) and (10) are equal, the equality of $b_{\mathrm{MW}}$ and $b_{C}$ is guaranteed. Therefore, if we equate the coefficients of $K_{1}{,\;K}_{2}$ and $k_{1}{,k}_{2}$ in (9) and (10) and compare the coefficients of $K_{1}{,\;K}_{2},$ we immediately have
(11)$$\beta_{K1}=\frac{f_{1}}{f_{1}{+f}_{2}}\;\mathrm{and}\;\beta_{K2}=\frac{f_{2}}{f_{1}{+f}_{2}}$$

Equating the coefficients of $k_{1}$ and $k_{2}\;$gives
(12)$$\frac{f_{1}^{2}\left(1-\beta_{k1}\right){+f}_{2}^{2}\beta_{k1}}{f_{1}^{2}-f_{2}^{2}}=\frac{f_{1}}{f_{1}-f_{2}}\Rightarrow \beta_{k1}=-\frac{f_{1}f_{2}}{f_{1}^{2}-f_{2}^{2}}=-\alpha_{w}$$
(13)$$\frac{f_{1}^{2}\beta_{k2}{+f}_{2}^{2}\left(1-\beta_{k2}\right)}{f_{1}^{2}-f_{2}^{2}}=\frac{f_{2}}{f_{1}-f_{2}}\Rightarrow \beta_{k2}=\frac{f_{1}f_{2}}{f_{1}^{2}-f_{2}^{2}}{=\alpha }_{w}$$

Therefore, instead of the conventional clock parameterization (1), we propose a novel parameterization which can be written as follows
(14)$$c\delta t=\mathrm{cdt}+\frac{f_{1}}{f_{1}{+f}_{2}}K_{1}+\frac{f_{2}}{f_{1}{+f}_{2}}K_{2}-\alpha_{w}k_{1}{+\alpha }_{w}k_{2}$$

With the clock parameterization (1), the receiver DCBs in the ionosphere-free code measurements in the standard ionosphere-free PPP approach are canceled out. The proposed parameterization (14) leads to the equality of the wide-lane and ionosphere-free biases $b_{\mathrm{MW}}$ and $b_{C}$. Below in Table 2, the biases for different dual-frequency carrier phase combinations with the proposed parameterization (14) are summarized, and their relations to their conventional counterparts are also given. For convenience, the carrier phase biases corresponding to the proposed parameterization are denoted as ${ϐ}_{*}$.

We now introduce a new bias ${ϐ}_{\mathrm{MWC}}$
(15)$${ϐ}_{\mathrm{MW}}{=ϐ}_{C}{=ϐ}_{\mathrm{MWC}}\equiv -\frac{f_{1}}{f_{1}{+f}_{2}}K_{1}-\frac{f_{2}}{f_{1}{+f}_{2}}K_{2}+\frac{f_{1}}{f_{1}-f_{2}}k_{1}-\frac{f_{2}}{f_{1}-f_{2}}k_{2}\;$$

It should be noted that throughout the paper we use the curlicue symbol ϐ to denote the carrier phase biases corresponding to the proposed parameterization. Inspection of the bias expressions given in Table 2 allows for making a few important conclusions: (1) the carrier phase bias ${ϐ}_{\mathrm{MWC}}$ is presented in the undifferenced measurements as well as in the ionosphere-free and the M-W combinations. So having computed ${ϐ}_{\mathrm{MWC}},$ we can easily take advantage of the mutual dependence of the biases ${ϐ}_{*}$ and are able to obtain either of them. Moreover, it is easy to see that the bias ${ϐ}_{\mathrm{MWC}}$ is the complete analog of the wide-lane bias $b_{W}$, which means that ${ϐ}_{\mathrm{MWC}}$ can be estimated in exactly the same way as $b_{W}$, namely from the M-W combination. As soon as ${ϐ}_{\mathrm{MWC}}$ is known, it can be in equal measure used in different PPP and PPP-RTK models based on single-frequency, ionosphere-free, or undifferenced multi-frequency carrier phase measurements. Additionally, (2) it is worth noticing that, unlike the user positioning models with the conventional clock parameterization, the wide-lane and M-W biases ${ϐ}_{\mathrm{MW}}\;$and ${ϐ}_{W}$ are different. This is why henceforth we have to distinguish between these two quantities and use subscript MW wherever necessary, e.g., in Equation (15). As a matter of fact, the condition of the equality of the wide-lane and ionosphere-free carrier phase biases should be written as ${ϐ}_{\mathrm{MW}}{=ϐ}_{C}$; (3) due to this condition and since the geometry-free bias ${ϐ}_{I}=\frac{1}{\alpha_{w}}\left({ϐ}_{\mathrm{MW}}-{ϐ}_{C}\right)$, it follows that ${ϐ}_{I}\equiv 0$, and, therefore, the geometry-free (float) bias $B_{I}$ becomes equal to its “integer” counterpart $\lambda_{1}N_{1}-\lambda_{2}N_{2}$. It should be mentioned here that some parallels with the PPP-AR products computed by CNES (Centre National d’Études Spatiales) IGS analysis center can be drawn. Namely, CNES provides the satellite with uncalibrated wide-lane phase delays obtained from the M-W combination [13] which may be considered an analog of ${ϐ}_{\mathrm{MWC}}$. Yet the definition of the integer clocks in the CNES PPP-AR product, apparently, implicates the employment of the standard ionosphere-free PPP model by the end-user.

## 3. Implications of the Proposed Clock Parametrization

### 3.1. Implication for Ionosphere Estimation 

New expressions for the uncombined carrier phase measurements at frequencies${\;f}_{1}$ and $f_{2}$ can be written as follows:(16)$$\Phi_{1}{=u}_{r}-{\tilde{\alpha }}_{1}\cdot I-\frac{f_{1}}{f_{1}{+f}_{2}}K_{1}-\frac{f_{2}}{f_{1}{+f}_{2}}K_{2}+\left({1+\alpha }_{w}\right)k_{1}-\alpha_{w}k_{2}{+\lambda }_{1}N_{1}{+\varepsilon }_{\Phi_{1}}{=u}_{r}-{\tilde{\alpha }}_{1}\left(I-k_{21}\right){+ϐ}_{\mathrm{MWC}}{+\lambda }_{1}N_{1}{+\varepsilon }_{\Phi_{1}}$$
(17)$$\Phi_{2}{=u}_{r}-{\tilde{\alpha }}_{2}\cdot I-\frac{f_{1}}{f_{1}{+f}_{2}}K_{1}-\frac{f_{2}}{f_{1}{+f}_{2}}K_{2}{+\alpha }_{w}k_{1}+\left(1-\alpha_{w}\right)k_{2}{+\lambda }_{2}N_{2}{+\varepsilon }_{\Phi_{2}}{=u}_{r}-{\tilde{\alpha }}_{2}\left(I-k_{21}\right){+ϐ}_{\mathrm{MWC}}{+\lambda }_{2}N_{2}{+\varepsilon }_{\Phi_{2}}$$
where $I=\left(\alpha_{2}-\alpha_{1}\right)\cdot \mathrm{STEC}$ and $\alpha_{i}=\frac{{40.3\cdot 10}^{16}}{f_{i}^{2}}$. The ionosphere-free and the M-W carrier phase combinations now read
(18)$$\Phi_{C}{=u}_{r}{+ϐ}_{\mathrm{MWC}}{+\lambda }_{N}\left(N_{1}+\frac{\lambda_{w}}{\lambda_{2}}N_{w}\right){+\varepsilon }_{\Phi_{C}}{=u}_{r}{+B}_{C}{+\varepsilon }_{\Phi_{C}}$$
(19)$$\Phi_{\mathrm{MW}}{=ϐ}_{\mathrm{MWC}}{+\lambda }_{w}N_{w}{+\varepsilon }_{\mathrm{MW}}{=B}_{\mathrm{MW}}{+\varepsilon }_{\mathrm{MW}}$$

Using Equations (16) and (17), the geometry-free carrier phase combination is
(20)$$\Phi_{I}{=\Phi }_{1}-\Phi_{2}=I+\left(\lambda_{1}N_{1}-\lambda_{2}N_{2}\right)-k_{21}{+\varepsilon }_{\Phi_{I}}{=I+B}_{I}-k_{21}{+\varepsilon }_{\Phi_{I}}$$

From Equations (18)–(20) we have
(21)$$\Phi_{\mathrm{MW}}-\Phi_{C}=-u_{r}{+B}_{\mathrm{MW}}-B_{C}+\epsilon =-u_{r}{+\alpha }_{w}\left(\lambda_{1}N_{1}-\lambda_{2}N_{2}\right)+\epsilon =-u_{r}{+\alpha }_{w}B_{I}+\epsilon =-u_{r}{+\alpha }_{w}\left(\Phi_{I}-{I+k}_{21}\right)+\epsilon$$
and, finally,
(22)$$\frac{1}{\alpha_{w}}\left(B_{\mathrm{MW}}-B_{C}\right){=B}_{I}{=\Phi }_{I}-{I+k}_{21}$$

This equation is the analog of Equation (2) used in the Fast-PPP model [3]. There is one essential difference. With the new parameterization (14), the slant ionosphere is no longer lumped with the DCB term $K_{21}$. Instead, it is biased by the considerably smaller magnitude term $k_{21}{=k}_{2}-k_{1}$, the inter-frequency bias between the carrier phase hardware delays at frequencies $f_{1}$ and $f_{2}$, or differential carrier phase bias (DPB). Here we make use of the fact that it is not necessary to know the true (absolute) hardware phase bias since the integer part can always be lumped with integer ambiguities, and so the remaining fractional (relative) part, which is then modulo 1 wavelength, can be provided as a carrier phase bias correction [14]. Therefore, given a cm-level magnitude of $k_{21}$, the difference between $B_{\mathrm{MW}}$ and $B_{C}$ can directly be used to retrieve accurate ionosphere information. Moreover, just one bias ${ϐ}_{\mathrm{MWC}}\;$is required to estimate $B_{\mathrm{MW}}\;$and $B_{C}\;$, and it can be easily estimated from the M-W combination as mentioned above.

### 3.2. Implication for User Positioning

Below we give a short summary of user positioning models that are associated with the proposed clock parameterization. A more detailed analysis is out of the scope of this publication and will be the topic for future research.

Let us consider the expressions for carrier phase and code measurements with the clock parameterization (14)
(23)$$\Phi_{1}{=u}_{r}-{\tilde{\alpha }}_{1}\left(I-k_{21}\right){+ϐ}_{\mathrm{MWC}}{+\lambda }_{1}N_{1}{+\varepsilon }_{\Phi_{1}}$$
(24)$$\Phi_{2}{=u}_{r}-{\tilde{\alpha }}_{2}\left(I-k_{21}\right){+ϐ}_{\mathrm{MWC}}{+\lambda }_{2}N_{2}{+\varepsilon }_{\Phi_{2}}$$
(25)$$R_{1}{=u}_{r}{+\tilde{\alpha }}_{1}\cdot I-\frac{f_{2}}{f_{1}{+f}_{2}}K_{21}-\alpha_{w}k_{21}{+\varepsilon }_{R_{1}}$$
(26)$$R_{2}{=u}_{r}+{\tilde{\alpha }}_{2}\cdot I+\frac{f_{1}}{f_{1}{+f}_{2}}K_{21}-\alpha_{w}k_{21}{+\varepsilon }_{R_{2}}.$$

To develop user positioning models, one needs to bear in mind that the code and carrier phase hardware delays, $K_{\left\{1,2\right\}}$ and $k_{\left\{1,2\right\}}$, are, in fact, the combined receiver and satellite delays: $K_{\left\{1,2\right\}}{=K}_{\left\{1,2\right\},r}-K_{\left\{1,2\right\}}^{s}$ and $k_{\left\{1,2\right\}}{=k}_{\left\{1,2\right\},r}-k_{\left\{1,2\right\}}^{s}$. Therefore, for the mathematical formulation of user positioning models, we need to decouple the satellite and receiver effects in the code and carrier phase biases and clock errors; the satellite part is provided as a part of correction information, the receiver part is lumped with the receiver clock to be estimated at the user end. In doing so, we arrive at the following expressions for the carrier phase and code measurements
(27)$$\Phi_{1}{=\rho }_{r}+c{\tilde{\delta t}}_{r}-{c\delta t}^{s}{+T}_{r}-{\tilde{\alpha }}_{1}\cdot \left({I+k}_{21}^{s}\right){+ϐ}_{\mathrm{MWC}}^{s}{+ϐ}_{\mathrm{MWC},r}{+\lambda }_{1}N_{1}{+\varepsilon }_{\Phi_{1}}$$
(28)$$\Phi_{2}{=\rho }_{r}+c{\tilde{\delta t}}_{r}-{c\delta t}^{s}{+T}_{r}-{\tilde{\alpha }}_{2}\cdot \left({I+k}_{21}^{s}\right){+ϐ}_{\mathrm{MWC}}^{s}{+ϐ}_{\mathrm{MWC},r}{+k}_{21,r}{+\lambda }_{2}N_{2}{+\varepsilon }_{\Phi_{2}}$$
(29)$$R_{1}{=\rho }_{r}+c{\tilde{\delta t}}_{r}-{c\delta t}^{s}{+T}_{r}+{\tilde{\alpha }}_{1}{\cdot I+\alpha }_{w}k_{21}^{s}-\frac{f_{2}}{f_{1}{+f}_{2}}\left(K_{21,r}-K_{21}^{s}\right)+\frac{f_{2}}{f_{1}-f_{2}}k_{21,r}{+\varepsilon }_{R_{1}}$$
(30)$$R_{2}{=\rho }_{r}+c{\tilde{\delta t}}_{r}-{c\delta t}^{s}{+T}_{r}+{\tilde{\alpha }}_{2}{\cdot I+\alpha }_{w}k_{21}^{s}+\frac{f_{1}}{f_{1}{+f}_{2}}\left(K_{21,r}-K_{21}^{s}\right)+\frac{f_{2}}{f_{1}-f_{2}}k_{21,r}{+\varepsilon }_{R_{2}}$$
where ${\tilde{\delta t}}_{r}{=\delta t}_{r}{+\tilde{\alpha }}_{1}k_{21,r}$.

As far as the receiver DPB term in the code measurements (29) and (30) is concerned, it can be neglected, since the magnitude of the DPBs, which is typically at a few cm level, is well below the code measurement noise. For the same reason, and because this term will mainly be absorbed by the receiver clock, the DPB term in (28) can be neglected as well. The satellite-specific bias ${ϐ}_{\mathrm{MWC}}^{s}$ is estimated and provided to the end-user as a part of correction information. As for its receiver counterpart, ${ϐ}_{\mathrm{MWC},r}$, it can be estimated from the M-W combination with ${ϐ}_{\mathrm{MWC}}^{s}$ accounted for beforehand. The ${ϐ}_{\mathrm{MWC},r}$ estimation uncertainty can be considered as an additional unknown and included in the state vector to be estimated by a navigation filter, in order to support ambiguity estimation and fixing. We analyzed this approach and below present the results of this analysis.

Inspection of Equations (29) and (30) reveals an elegant way to discard both the satellite and receiver DCBs. It is seen that the DCB terms have coefficients $-\frac{f_{2}}{f_{1}{+f}_{2}}$ and $+\frac{f_{1}}{f_{1}{+f}_{2}}$ for the L1 and L2 code measurements, respectively. This means that in the narrow-lane combination of the code measurements, these terms cancel out, and thus
(31)$$R_{n}=\frac{f_{1}}{f_{1}{+f}_{2}}R_{1}+\frac{f_{2}}{f_{1}{+f}_{2}}R_{2}{+\varepsilon }_{n}{=\rho }_{r}+c{\tilde{\delta t}}_{r}-{c\delta t}^{s}{+T}_{r}{+\alpha }_{w}{\cdot I+\alpha }_{w}k_{21}^{s}+\frac{f_{2}}{f_{1}-f_{2}}k_{21,r}{+\varepsilon }_{n}$$

The undifferenced carrier phases given in Equations (27) and (28) together with the narrow-lane code combination (31) can be considered as a dual-frequency PPP-RTK user positioning model exploiting the proposed parameterization (14). It can be summarized as follows
(32)$$\Phi_{1}{=\rho }_{r}+c{\tilde{\delta t}}_{r}-{c\delta t}^{s}{+T}_{r}-{\tilde{\alpha }}_{1}\cdot \left({I+k}_{21}^{s}\right){+ϐ}_{\mathrm{MWC}}^{s}{+ϐ}_{\mathrm{MWC},r}{+\lambda }_{1}N_{1}{+\varepsilon }_{\Phi_{1}}$$
(33)$$\Phi_{2}{=\rho }_{r}+c{\tilde{\delta t}}_{r}-{c\delta t}^{s}{+T}_{r}-{\tilde{\alpha }}_{2}\cdot \left({I+k}_{21}^{s}\right){+ϐ}_{\mathrm{MWC}}^{s}{+ϐ}_{\mathrm{MWC},r}{+\lambda }_{2}N_{2}{+\varepsilon }_{\Phi_{2}}$$
(34)$$R_{n}{=\rho }_{r}+c{\tilde{\delta t}}_{r}-{c\delta t}^{s}{+T}_{r}{+\alpha }_{w}{\cdot I+\alpha }_{w}k_{21}^{s}{+\varepsilon }_{n}$$

An advantage of using such a mathematical model for user positioning, which will be referred to as an undifferenced positioning model, is that it is not necessary to cope with code hardware delays. The only information provided to the end-user in this case is the satellite clock, troposphere and ionosphere delays, the satellite DPB, and the satellite-dependent bias ${ϐ}_{\mathrm{MWC}}^{s}$. The necessity to estimate one extra parameter for the carrier phases can be considered a drawback of the model.

The undifferenced model can easily be extended to the multi-constellation case. If we introduce the designation Ҁ for different GNSS systems Ҁ: G=GPS,E=GALILEO,C=BEIDOU,J=QZSS thus, for the sake of simplicity, considering only systems transmitting CDMA signals (therefore, excluding GLONASS), we can re-write Equations (32)–(34) as follows
(35)$$\Phi_{1}^{\left(Ҁ\right)}{=\rho }_{r}+c{\tilde{\delta t}}_{r}^{\left(Ҁ\right)}-{c\delta t}^{s}{+T}_{r}-{\tilde{\alpha }}_{1}^{\left(Ҁ\right)}\cdot \left({I+k}_{21}^{s}\right){+ϐ}_{\mathrm{MWC}}^{s}{+ϐ}_{\mathrm{MWC},r}^{\left(Ҁ\right)}{+\lambda }_{1}^{\left(Ҁ\right)}N_{1}^{\left(Ҁ\right)}{+\varepsilon }_{\Phi_{1}}^{\left(Ҁ\right)}$$
(36)$$\Phi_{2}^{\left(Ҁ\right)}{=\rho }_{r}+c{\tilde{\delta t}}_{r}^{\left(Ҁ\right)}-{c\delta t}^{s}{+T}_{r}-{\tilde{\alpha }}_{2}^{\left(Ҁ\right)}\cdot \left({I+k}_{21}^{s}\right){+ϐ}_{\mathrm{MWC}}^{s}{+ϐ}_{\mathrm{MWC},r}^{\left(Ҁ\right)}{+\lambda }_{2}^{\left(Ҁ\right)}N_{2}^{\left(Ҁ\right)}{+\varepsilon }_{\Phi_{2}}^{\left(Ҁ\right)}$$
(37)$$R_{n}^{\left(Ҁ\right)}{=\rho }_{r}+c{\tilde{\delta t}}_{r}^{\left(Ҁ\right)}-{c\delta t}^{s}{+T}_{r}{+\alpha }_{w}^{\left(Ҁ\right)}{\cdot I+\alpha }_{w}^{\left(Ҁ\right)}k_{21}^{s}{+\varepsilon }_{n}^{\left(Ҁ\right)}$$
with ${\tilde{\delta t}}_{r}^{\left(Ҁ\right)}{=\delta t}_{r}^{\left(Ҁ\right)}+{\tilde{\alpha }}_{1}^{\left(Ҁ\right)}k_{21,r}^{\left(Ҁ\right)}$. Again, the receiver-specific bias ${ϐ}_{\mathrm{MWC},r}$ is estimated from the M-W combination and, in order to support ambiguity estimation and fixing, we introduce an extra parameter to be estimated. As a preliminary step, and to avoid estimating too many new unknowns by a Kalman Filter, this parameter is assumed to be GNSS-dependent. A more detailed investigation of possible approaches to estimate the receiver part of the bias ${ϐ}_{\mathrm{MWC},r}$ as well as their impact on the ambiguity resolution performance is out of the scope of this publication and is an interesting topic for future works. Finally, if we add FDMA signals to this model, we have to additionally take into consideration the inter-frequency code and carrier phase biases.

## 4. Referring Higher Frequency Measurements to Re-Parameterized Clocks 

Here, we briefly consider an important topic of how to extend the proposed dual-frequency user positioning models to the multi-GNSS case. We use a quite obvious approach of expressing third and higher frequency measurements in terms of the re-parameterized according to (14) clock δt. For triple-frequency user positioning models we need to replenish Equations (23)–(26) with additional expressions for code and carrier phases at frequency $f_{3}$
(38)$$\Phi_{3}{=u}_{r}-{\tilde{\alpha }}_{3}\cdot I-{\tilde{\alpha }}_{2}k_{1}+{\tilde{\alpha }}_{1}k_{2}{+k}_{3}{+ϐ}_{\mathrm{MWC}}{+\lambda }_{3}N_{3}{+\varepsilon }_{\Phi_{3}}$$
(39)$$R_{3}{=u}_{r}+{\tilde{\alpha }}_{3}\cdot I-\frac{f_{2}}{f_{1}{+f}_{2}}K_{21}{+K}_{31}-\alpha_{w}k_{21}{+\varepsilon }_{R_{3}}$$
where ${\tilde{\alpha }}_{3}=\frac{f_{1}^{2}f_{2}^{2}}{f_{3}^{2}}\frac{1}{f_{1}^{2}-f_{2}^{2}}$ and $K_{31}{=K}_{3}\;-{\;K}_{1}$. Now, we separate the satellite and receiver parts of the bias ${ϐ}_{\mathrm{MWC}}$ as well as the code and carrier phase differential biases
(40)$$\Phi_{3}{=\rho }_{r}+c{\tilde{\delta t}}_{r}-{c\delta t}^{s}{+T}_{r}-{\tilde{\alpha }}_{3}\cdot I-{\tilde{\alpha }}_{2}k_{1}^{s}+{\tilde{\alpha }}_{1}k_{2}^{s}{+ϐ}_{\mathrm{MWC}}^{s}{+ϐ}_{\mathrm{MWC},r}{+k}_{31,r}{+\lambda }_{3}N_{3}{+\varepsilon }_{\Phi_{3}}$$
(41)$$R_{3}{=\rho }_{r}+c{\tilde{\delta t}}_{r}-{c\delta t}^{s}{+T}_{r}+{\tilde{\alpha }}_{3}\cdot I-\alpha_{w}k_{21}^{s}-\frac{f_{2}}{f_{1}{+f}_{2}}\left(K_{21,r}-K_{21}^{s}\right)+\left(K_{31,r}-K_{31}^{s}\right)-\frac{f_{2}}{f_{1}-f_{2}}k_{21,r}{+\varepsilon }_{R_{3}}$$

In general, we can write down the corresponding expressions at arbitrary frequency ν ≥ 3
(42)$$\Phi_{\nu }{=\rho }_{r}+c{\tilde{\delta t}}_{r}-{c\delta t}^{s}{+T}_{r}-{\tilde{\alpha }}_{\nu }\cdot I-{\tilde{\alpha }}_{2}k_{1}^{s}+{\tilde{\alpha }}_{1}k_{2}^{s}{+ϐ}_{\mathrm{MWC}}^{s}{+ϐ}_{\mathrm{MWC},r}{+k}_{\nu 1,r}{+\lambda }_{\nu }N_{\nu }{+\varepsilon }_{\Phi_{\nu }}$$
(43)$$R_{\nu }{=\rho }_{r}+c{\tilde{\delta t}}_{r}-{c\delta t}^{s}{+T}_{r}+{\tilde{\alpha }}_{\nu }\cdot I-\alpha_{w}k_{21}^{s}-\frac{f_{2}}{f_{1}{+f}_{2}}\left(K_{21,r}-K_{21}^{s}\right)+\left(K_{\nu 1,r}-K_{\nu 1}^{s}\right)-\frac{f_{2}}{f_{1}-f_{2}}k_{21,r}{+\varepsilon }_{R_{\nu }}$$
where ${\tilde{\alpha }}_{\nu }=\frac{f_{1}^{2}f_{2}^{2}}{f_{\nu }^{2}}\frac{1}{f_{1}^{2}-f_{2}^{2}}$. Again, as can be seen from Equations (29), (30), (39) and (41), the DCB terms have coefficients $-\frac{f_{2}}{f_{1}{+f}_{2}}$ and $+\frac{f_{1}}{f_{1}{+f}_{2}}$ for $L_{\nu =3,\cdots }$ and the $L_{2}$ code measurements, respectively, so in the following narrow-lane combinations these terms cancel out
(44)$$R_{\nu 2,n}=\frac{f_{1}}{f_{1}{+f}_{2}}R_{\nu }+\frac{f_{2}}{f_{1}{+f}_{2}}R_{2}{=\rho }_{r}+c{\tilde{\delta t}}_{r}-{c\delta t}^{s}{+T}_{r}+{\tilde{\alpha }}_{n}\cdot I-\alpha_{w}k_{21}^{s}+\frac{f_{1}}{f_{1}{+f}_{2}}\left(K_{\nu 1,r}-K_{\nu 1}^{s}\right)-\alpha_{w}k_{21,r}{+\varepsilon }_{\nu 2,n}.$$
where ${\tilde{\alpha }}_{n}={\tilde{\alpha }}_{\nu }\frac{f_{1}}{f_{1}{+f}_{2}}+{\tilde{\alpha }}_{2}\frac{f_{2}}{f_{1}{+f}_{2}}$. Now, the receiver DPB term in the code measurements (44) can be neglected since its magnitude is well below the code measurement noise. Therefore, the following multi-frequency model can be employed for user positioning
(45)$$\Phi_{1}{=\rho }_{r}+c{\tilde{\delta t}}_{r}-{c\delta t}^{s}{+T}_{r}-{\tilde{\alpha }}_{1}\cdot \left({I+k}_{21}^{s}\right){+ϐ}_{\mathrm{MWC}}^{s}{+ϐ}_{\mathrm{MWC},r}{+\lambda }_{1}N_{1}{+\varepsilon }_{\Phi_{1}}$$
(46)$$\Phi_{2}{=\rho }_{r}+c{\tilde{\delta t}}_{r}-{c\delta t}^{s}{+T}_{r}-{\tilde{\alpha }}_{2}\cdot \left({I+k}_{21}^{s}\right){+ϐ}_{\mathrm{MWC}}^{s}{+ϐ}_{\mathrm{MWC},r}{+\lambda }_{2}N_{2}{+\varepsilon }_{\Phi_{2}}$$
(47)$$\Phi_{\nu }{=\rho }_{r}+c{\tilde{\delta t}}_{r}-{c\delta t}^{s}{+T}_{r}-{\tilde{\alpha }}_{\nu }\cdot I-{\tilde{\alpha }}_{2}k_{1}^{s}+{\tilde{\alpha }}_{1}k_{2}^{s}{+ϐ}_{\mathrm{MWC}}^{s}{+ϐ}_{\mathrm{MWC},r}{+k}_{\nu 1,r}{+\lambda }_{\nu }N_{\nu }{+\varepsilon }_{\Phi_{\nu }}$$
(48)$$R_{12,n}{=\rho }_{r}+c{\tilde{\delta t}}_{r}-{c\delta t}^{s}{+T}_{r}{+\alpha }_{w}{\cdot I+\alpha }_{w}k_{21}^{s}{+\varepsilon }_{n}$$
(49)$$R_{\nu 2,n}{=\rho }_{r}+c{\tilde{\delta t}}_{r}-{c\delta t}^{s}{+T}_{r}+{\tilde{\alpha }}_{n}\cdot I-\alpha_{w}k_{21}^{s}+\frac{f_{1}}{f_{1}{+f}_{2}}\left(K_{\nu 1,r}-K_{\nu 1}^{s}\right){+\varepsilon }_{\nu 2,n}.$$
where $R_{*2,n}$ is the narrow-lane combination of the code measurements for frequencies $f_{*}{=f}_{\left\{1,3,\cdots \right\}}$, and frequency $f_{2}$ with the coefficients $\frac{f_{1}}{f_{1}{+f}_{2}}$ and $\frac{f_{2}}{f_{1}{+f}_{2}}$. We see that, unlike in the dual-frequency case, in the multi-frequency positioning model we have to provide to the user the satellite DCBs for frequencies ν≥3. Moreover, their receiver counterparts have to be estimated as well. This circumstance is a weak point of the multi-frequency positioning model introduced above.

## 5. Results

### 5.1. Data Processing and Corrections Generation 

We assessed the dual-frequency user positioning model given by Equations (27), (28), and (31). For data processing, we used a modified version of an open-source PPP-RTK positioning toolkit CLASLIB [15]. The code was modified to implement the data processing strategy described below.

In order to demonstrate the performance under practical circumstances, data processing was carried out in two stages. In the first stage, the data from a global network of stations were processed and analyzed to generate a set of corrections. This primarily concerned the ionosphere slant delays, and the bias ${ϐ}_{\mathrm{MWC}}^{s}$, and the DPBs, since the accurate satellite state information is taken from the final orbit and clock products by the CODE IGS Analysis Center [16]. We considered daily measurements from 60 IGS stations covering the globe as uniformly as possible. The stations were chosen arbitrarily, as long as they provided enough information to generate correction information. The distribution of the stations is shown in Figure 1.

In the second stage, the corrections generated were applied to measurements of a static receiver using the undifferenced user positioning model given by Equations (27), (28), and (31). A summary of the data processing strategy is presented in Table 3.

The algorithm for generating the corrections was performed on an epoch-by-epoch basis and can be briefly described as follows:
The differential phase bias $k_{21}$ and the ionospheric slant delay I were evaluated using the geometry-free combination of carrier phases. At the first epoch, pairs of ambiguities $N_{1}$ and $N_{2}$ were sought such that their geometry-free combination matched as close as possible the geometry-free measurements. These ambiguities were held fixed at the subsequent epochs, so far as no cycle slips occur. If a cycle slip was detected by using the M-W and Geometry-Free combinations, its size was determined and the ambiguities $N_{1}$ and $N_{2}$ were corrected accordingly. The remaining non-zero part was split into the constant ($k_{21}$) and the time-varying (∆I) parts. The latter was assumed to be zero at the first epoch, so STEC at the first epoch equaled the corresponding IGS GIM ionosphere estimate and at the subsequent epochs was reconstructed as GIM ionosphere estimates at the initial epoch plus ∆I evaluated from the carrier phases.Once $N_{1}$ and $N_{2}$ were fixed, we could find $N_{\mathrm{WL}}$ and evaluate the bias ${ϐ}_{\mathrm{MWC}}$ from the M-W combination.Having $N_{1}$, I, $k_{21}$, ${ϐ}_{\mathrm{MWC}}\;$available, the evaluation was adjusted to the new parameterization clock error ${\delta t}_{r}^{s}$ from carrier phases on L1.With I, we evaluated code DCB $K_{21}$ from the geometry-free code measurements.

A graphical representation of this algorithm is given in Figure 2.

It is worth noticing here that our goal was primarily to demonstrate the proposed clock parameterization and, specifically, to verify the correctness of the corresponding PPP-RTK models, so we intended to develop and implement a simple algorithm to derive a set of corrections preserving the integer nature of the ambiguities, which is necessary for the user position accuracy and the ambiguity resolution assessment. This is why we began directly with fixing the ambiguities to integers as explained above. Moreover, we imposed additional constraints by employing a-priori information about the ionosphere (GIM values) and the satellite states (orbits and clocks), see Table 3. By doing so, we have removed the necessity to deal with rank deficiency resolution and the assessment of the estimability of the corrections, which is necessary for models with undifferenced GNSS observation Equations [22]. Additionally, direct setting ambiguities to integers and constraining the ionosphere not only simplifies the algorithm but also allows obtaining an internally consistent set of corrections with the unbiased ionosphere information. It is to be noted here that this algorithm of correction generation is only experimental, and merely aimed at providing necessary information to be able to demonstrate the practical use of the afore-introduced user positioning models. This is why it will not be considered in more detail in this contribution. In the future, we will continue working on the algorithm towards a more rigorous derivation of an estimable set of corrections from a global network of receivers for the models with the proposed parameterization.

The algorithm provides only the differential code and carrier phase biases that combine the receiver- and satellite-dependent effects on two frequencies. Additional steps are required to separate these effects and arrive at the satellite-specific biases. These steps can be summarized as follows:
Obtain $k_{1}$, $k_{2}$, $K_{1}$, $K_{2}$ from $k_{21}$ and $K_{21}$;Separate the receiver- and satellite-specific parts of the code and carrier phase hardware biases $k_{\left\{1,2\right\}}\to k_{\{1,2\},r}{,k}_{\left\{1,2\right\}}^{s}$ and $K_{\left\{1,2\right\}}\to K_{\{1,2\},r}{,K}_{\left\{1,2\right\}}^{s}$;Use $k_{\left\{1,2\right\}}^{s}$ and $K_{\left\{1,2\right\}}^{s}$ to compute $k_{21}^{s}$ and ${ϐ}_{\mathrm{MWC}}$.

Visualization of this additional bias generation algorithm is shown in Figure 3.

To separate receiver- and satellite-specific hardware biases, we applied the Least Squares-based approach described in [23]. Finally, the bias ${ϐ}_{\mathrm{MWC}}^{s}$, the DPBs $k_{21}^{s},$ along with the ionosphere slant delay I, and the satellite clock and orbit estimates are subsequently applied at the user positioning stage.

Despite the demonstrational character of the correction generation algorithm, this set of corrections is associated with the PPP-RTK model considered in this contribution and is to be provided whatever algorithm is used for its computation. An important question of how to connect the proposed model to the PPP-RTK methods, which were formulated in past years [24,25,26,27], needs to be posed. To answer it, a rigorous and detailed comparative analysis of the afore-described model is required. This will be the subject for future work, in this contribution we only outline two possible approaches to perform such an analysis. One of the possibilities is to employ the PPP-AR products interoperability concept proposed in [28], which is based on the Observable-Specific signal Bias (OSB) approach [29]. As a starting point, in order to give the reader a better vision of this possibility, we demonstrate below how our clock and phase bias corrections could have been represented as functions of OSBs in matrix form, assuming that C1W and C2W signals are tracked and retaining the corresponding notation from [28]:(50)$$\left[\begin{array}{c} {\tilde{dt}}^{j}\\ {\mathrm{DCB}}_{C1W,C2W}\\ {ϐ}_{\mathrm{MWC}}^{j} \end{array}\right]=\left[\begin{array}{ccccc} 1 & -\alpha_{\mathrm{NL}} & \alpha_{\mathrm{NL}}-1 & \alpha_{w} & -\alpha_{w}\\ 0 & 1 & -1 & 0 & 0\\ 0 & -\alpha_{\mathrm{NL}} & -\beta_{\mathrm{NL}} & \alpha_{\mathrm{WL}} & \beta_{\mathrm{WL}} \end{array}\right]\left[\begin{array}{c} \begin{array}{c} {\mathrm{dt}}^{j}\\ {\tilde{b}}_{C1W}\\ {\tilde{b}}_{C2W}\\ {\tilde{b}}_{L1} \end{array}\\ {\tilde{b}}_{L2} \end{array}\right]$$
where $\alpha_{\mathrm{WL}}=\frac{f_{1}}{f_{1}-f_{2}}$, $\alpha_{\mathrm{NL}}=\frac{f_{1}}{f_{1}{+f}_{2}}$, $\beta_{*}=1-\alpha_{*}$. For the sake of simplicity, we have omitted the analysis center timing offset term as well as the additional code bias constraint herein.

The second possibility is to take advantage of the results of the study demonstrated in [30] linking various PPP-RTK methods by using the S-system theory. As pointed out there, the interpretation of the estimable parameters is a key factor to gain a proper understanding of the role of the corresponding PPP-RTK corrections.

### 5.2. Static User Positioning

Each of the 60 IGS stations selected and shown in Figure 1 plays the role of the rover. The daily time spans for each station were divided into eight 3 h-long sessions which were processed one by one as if in real-time. The rover positions were estimated by a Kalman filter with the afore-described corrections applied to the proposed dual-frequency user positioning model given by Equations (27), (28), and (31). In total, we have 480 3 h-long position time series allowing us to study the performance in terms of the positioning accuracy, the convergence, and the ambiguity resolution performance. Three different schemes depending on how the user handles the bias ${ϐ}_{\mathrm{MWC}}^{s}$ are considered. Scheme 1 assumes that ${ϐ}_{\mathrm{MWC}}$ is entirely derived, epoch-wise, from the M-W combination, and so ${ϐ}_{\mathrm{MWC}}^{s}$ from the corrections is, in fact, not applied. In our validation, this scheme is used to check the mathematical correctness of the model proposed. Scheme 2 assumes that the user does apply ${ϐ}_{\mathrm{MWC}}^{s}$ and derives the receiver-specific counterpart ${ϐ}_{\mathrm{MWC},r}$ from the M-W combination at each epoch. Moreover, we introduced a new GNSS system-dependent parameter to the state vector that comprises the uncertainty of the receiver-specific bias ${ϐ}_{\mathrm{MWC},r}$ estimation using the M-W combination. The estimated bias ${ϐ}_{\mathrm{MWC},r}$ is used at each observation epoch as an initial value defined as the median over all satellites for a given GNSS system. The new GNSS system-dependent parameter is modeled as white noise. Scheme 3 essentially replicates Scheme 2 except that the ${ϐ}_{\mathrm{MWC},r}$ estimated from the M-W combination is used as an initial value at the first observation epoch only. Scheme 2 and Scheme 3 are considered to assess the effect of the introduction of a new GNSS-dependent parameter on the user positioning performance.

Figure 4 shows the distribution of the RMS positioning errors with respect to the precise ITRF14 coordinates obtained with the proposed dual-frequency PPP-RTK model. It is seen that for the overwhelming majority of the cases the errors are within 1 cm horizontal and 5 cm vertical for Scheme 1. Scheme 1 represents a somewhat idealized case when at each observation epoch the value of the bias ${ϐ}_{\mathrm{MWC}}$, pre-computed at the correction generation stage, is available. In practice, the interval estimate averaged over some time will be provided. This scheme demonstrates, therefore, the expected accuracy achievable with the undifferenced model given by Equations (27), (28), and (31) and brings some evidence of the correctness of the mathematical model and its implementation. Interestingly, if we introduce and estimate an additional GNSS-dependent parameter that uses the satellite-specific part ${ϐ}_{\mathrm{MWC}}^{s}$ as an initial value (Scheme 2), we observe only a marginal difference compared with the results for Scheme 1. In this case, initialization occurs at each observation epoch. However, if we initialize the additional parameter at the first observation epoch only (Scheme 3), the results become slightly worse, although they remain at a few cm agreement level.

We then analyzed the convergence of the static position estimates. We assumed that the convergence is achieved when for five consecutive epochs the absolute position error is below 10 cm. The cumulative histograms shown in Figure 5 demonstrate an instant position convergence for both the horizontal and vertical components for Scheme 1 for all the time series considered. This result is expected, since at the network processing stage we deliberately set all ambiguities to integers and derive the Differential Phase Biases $k_{21}$ and the bias ${ϐ}_{\mathrm{MWC}}$ based on this choice, such that these biases are associated with these known integers. For Scheme 2 the convergence times are only marginally different from that for Scheme 1. As far as Scheme 3 is concerned, nearly instant convergence (a few 30-s epochs) is only achieved for the horizontal component and a slightly smaller number of the cases compared with Scheme 1 and Scheme 2. Essential degradation in the vertical convergence is observed. Typical values range from a few up to several tens of 30-s epochs.

To conduct ambiguity resolution, we applied the Partial Ambiguity Resolution strategy implemented in the LAMBDA software [31]. For the validation of the ambiguity resolution results, the resolved ambiguities are compared with the reference ambiguities known from the network processing stage. The empirical and the bootstrapped success rates are employed as validation criteria. We use the following definition for the empirical success rate [32]:(51)$$P_{s}=\frac{\text{\#}\mathrm{correctly}\;\mathrm{fixed}\;\mathrm{ambiguities}}{\text{\#}\mathrm{total}\;\mathrm{ambiguities}}$$

Moreover, the bootstrapped success rate, which is a sharp lower bound of the integer least squares success rate [33,34], is computed and compared with the user-defined threshold of 99.5%. Figure 6 represents the results of the ambiguity resolution validation shown as the cumulative success rates. Again, the results are very similar for Scheme 1 and Scheme 2, which are both superior to the Scheme 3 ambiguity resolution results. As is seen from the plot on the left-hand side, the number of correctly fixed ambiguities was typically as high as 90–100% for the first two schemes, whereas Scheme 3 demonstrated the number of correctly fixed ambiguities at the level of 30–50% only. At the same time, the results on the right-hand side plot show the bootstrapped success rate of nearly 100% for all the Schemes, providing some evidence that the estimated integer ambiguities in all three cases coincided with the correct integer values with a very high level of statistical confidence.

Therefore, the discussed results bring some evidence of the feasibility of estimating the receiver-specific bias ${ϐ}_{\mathrm{MWC},r}$ at each observation epoch from the M-W combination and subsequently, combine it with its provided satellite-specific counterpart. Uncertainty in the estimates of the bias ${ϐ}_{\mathrm{MWC},r}$ may be compensated by adding one GNSS-dependent parameter to a Kalman filter state vector without essential loss of performance (as demonstrated by the results for Scheme 1 and Scheme 2). It is not sufficient to estimate the receiver-specific bias ${ϐ}_{\mathrm{MWC},r}$ only at the first observation epoch and use it to initialize the parameter estimation process, since this leads to noticeable degradation of the performance for the positioning accuracy, convergence, and ambiguity resolution.

### 5.3. Stability Analysis of Bias ${ϐ}_{\mathrm{MWC}}^{s}$

As follows from the above, the bias ${ϐ}_{\mathrm{MWC}}$ plays an important role in the positioning models associated with the proposed clock parameterization (14). In particular, its satellite-specific component ${ϐ}_{\mathrm{MWC}}^{s}$ is supposed to be provided as a part of the correction information. The different ways of handling the bias ${ϐ}_{\mathrm{MWC}}$ were discussed in the previous subsection. Below, we demonstrate the results of the long- and short-time stability analysis of this bias. For the long-time stability analysis, we processed the observation data from a network of the IGS stations shown in Figure 1 for 16 consecutive days from DOY50 to DOY65, in the year 2020. Single epoch estimates of ${ϐ}_{\mathrm{MWC}}^{s}$ are averaged out over a 1-day time span. The resulting mean values are presented in Figure 7 for GPS (a) and GALILEO (b) for carrier phases at frequency bands L1 and E1, respectively. We used the RINEX observations code, so “1C” corresponds to L1 C/A for GPS and E1 pilot channel for GALILEO. For better visualization, the bias time series for individual satellites are shifted along the y-axis by (PRN−16) × 0.1.

It is seen that the daily estimates demonstrate quite noticeable stability. It stems from the similarity between ${ϐ}_{\mathrm{MWC}}^{s}$ and the wide-lane bias $b_{w}$, see Table 2, and from the fact that $b_{w}$ is characterized by a stable behavior. The GALILEO biases demonstrate somewhat more significant variability compared with GPS. This can be explained by a smaller number of GALILEO satellites visible and, as a result, a smaller amount of measurement information available. This phenomenon can clearly be seen in Figure 8, giving the distributions of the residuals of the GPS and GALILEO daily ${ϐ}_{\mathrm{MWC}}^{s}$ estimates with respect to the overall 16-day mean. On the plot, the approximating normal distributions with the expectation μ and the standard deviation σ, the mode ν, and the percentage of the residuals below a specified threshold in the absolute magnitude are displayed.

In order to analyze the short-term stability, we have averaged out the single epoch estimates of ${ϐ}_{\mathrm{MWC}}^{s}$ over 3 h- and 1 h-long time spans. Doing so, we bear in mind that, in practice, the biases will not be available at each epoch of observations, but rather assumed constants over a certain time interval and updated at the end of such interval. Therefore, we compare two cases when the biases are updated every 3 and 1 h. The corresponding sub-daily bias estimates are presented in Figure 9 and Figure 10, respectively. Again, for better visualization, the bias time series for individual satellites are shifted along the y-axis by (PRN−16) × 0.25.

We can see that, in general, the bias estimates for both the constellations are noticeably stable, similar to their daily counterparts. At the same time, as in the case of the daily estimates, the GALILEO bias estimates reveal a higher variability compared with their GPS counterparts.

We then compare distributions of the residuals of the sub-daily (1 h and 3 h long) estimates of ${ϐ}_{\mathrm{MWC}}^{s}$, in order to find out which bias update time step may be more optimal. The distributions are demonstrated in Figure 11.

On the plot, the approximating normal distributions with the expectation μ and the standard deviation σ, the mode ν, and the percentage of the residuals below a specified threshold in the absolute magnitude are displayed. It can be noted that more than 90% of the bias residuals are within 0.2 cycles in absolute magnitude. This confirms the conclusions made above regarding the stability of the sub-daily bias estimates. In addition, it is clearly seen that the approximating normal distributions are very similar. This brings some evidence that the two samples under consideration were drawn from the same distribution and there is no statistical difference between the bias estimates over 1 h and 3 h time intervals. This means that apparently, it is sufficient to have bias updates every 3 h, and there no need to shorten it, although the problem of finding the most optimal update interval for ${ϐ}_{\mathrm{MWC}}^{s}$ requires more careful analysis. This, however, is out of the scope of this contribution and will be a topic for future research.

### 5.4. Retrieval of Ionosphere Slant Delays

Above, we derived Equation (22) linking the biases $B_{\mathrm{MW}}$, $B_{I}$, and $B_{C}$ similar to the one used in the framework of the Fast-PPP approach. With the proposed clock formulation, such equation, which will be referred to as the “ionosphere retrieval equation” throughout the rest of this work, provides ionosphere slant delays biased only by the carrier phase DPB, which opens up an opportunity to directly retrieve accurate slant ionosphere estimates. Below, we analyze this problem and provide some results demonstrating the performance of ionosphere retrieval using the “ionosphere retrieval Equation” (22). We considered the same network of stations shown in Figure 1 and processed daily measurements for DOY50, in the year 2020. We took the results of user positioning, namely, epoch-wise successfully fixed dual-frequency ambiguities $N_{1}$ and $N_{2}$. These were used to form the biases $B_{\mathrm{MW}}$, $B_{I}$, and $B_{C}$. Subsequently, Equation (22) was applied to derive slant ionosphere delays. The carrier phase DCB term $k_{12}$ was neglected. We then compared the resulting ionosphere slant delays, which will be referred to as “aPPP”—“alternative PPP”—in this work, with the Global Ionosphere Map (GIM) VTEC values from different Ionosphere Associate Analysis Centers (IAAC). A list of the GIMs used in the analysis is presented in Table 4.

We compare the solutions over the 24-h time span in terms of the average (bias) and Standard Deviation of the differences [35]:(52)$${\mathrm{Bias}}_{\mathrm{aPPP}\text{-}\mathrm{GIM}}=\frac{1}{N}\overset{N}{\underset{i=1}{\sum }}\left({\mathrm{VTEC}}_{\mathrm{aPPP}}^{i}-{\mathrm{VTEC}}_{\mathrm{GIM}}^{i}\right)$$
(53)$${\mathrm{STD}}_{\mathrm{aPPP}\text{-}\mathrm{GIM}}=\sqrt{\frac{\sum_{i=1}^{N}{\left({\mathrm{VTEC}}_{\mathrm{aPPP}}^{i}-{\mathrm{VTEC}}_{\mathrm{GIM}}^{i}-{\mathrm{Bias}}_{\mathrm{aPPP}\text{-}\mathrm{GIM}}\right)}^{2}}{N-1}}$$

Conversion of VTEC into STEC is performed by means of the Modified Single Layer Mapping function (MSLM) [36]
(54)$$F\left(z\right)=\frac{1}{\sqrt{1-{\left(\frac{R}{R+H}\sin\left(\alpha \cdot z\right)\right)}^{2}}}$$
where z is the zenith distance at the ionospheric shell of maximum ionization and the parameters R = 6371 km, H = 506.7 km, and α = 0.9782. The MSLM function is widely used since it delivers the best fit to the extended slab mapping function employed by JPL [37].

Below, in Figure 12 and Figure 13, some typical examples of STEC values derived from the ionosphere retrieval equation (model “aPPP”) are displayed together with the GIM STECs from the six IAACs. As is seen, the solutions demonstrate a noticeable mutual agreement between all the solutions at a level of several tenths of TECU, also preserved for low elevation angles. At the same time, there are a number of cases, which mostly occurred for the stations at low latitudes, when the aPPP STEC values overestimate the GIM estimates as can be seen from Figure 14. These examples clearly show that the differences between aPPP and GIM STECs can reach up to several TECUs.

If we take a look at the overall statistics of the aPPP-GIM VTEC differences, shown in Figure 15 and Figure 16 for GPS and GALILEO, respectively, we can find out that an agreement of 1–2 TECU with the standard deviation of 3–4 TECU can be reached. Interestingly, the mean values μ are all positive, showing that the impact of the overestimating cases shown in Figure 14 on the resulting statistics is quite significant. It causes large magnitudes of minimum and maximum differences, also displayed in Figure 15 and Figure 16. The origin of this phenomenon is unclear and may be attributed to the relative difficulty of accurate modeling of ionospheric parameters at low latitudes due to the presence of low latitude ionospheric phenomena.

Therefore, the results demonstrated provide some evidence of the good potential of the ionosphere retrieval equation derived above. It is worth noticing here that the results are of a tentative character. Here, we only confine ourselves to a demonstration of the general applicability and usability of the models associated with the proposed clock formulation to accurate ionosphere retrieval. To arrive at more firm conclusions, a more detailed and in-depth investigation is necessary.

## 6. Summary and Conclusions

In the presented work, we proposed and analyzed a novel clock parameterization which is different from the conventional formulation referring clocks to the L3 phase center and used in the IGS orbit and clock solutions. The proposed parameterization (14) leads to the equality of the wide-lane and ionosphere-free biases $b_{\mathrm{MW}}$ and $b_{C}$ and we considered the most important implications that follow from this property of the introduced parameterization. First, we derived the PPP-RTK user positioning models for dual-frequency code and carrier phase measurements for the undifferenced measurements. It was additionally proposed to employ the narrow-lane code combination instead of the undifferenced code measurements. It was then shown how this model can be extended to the third- and higher-frequency cases. Second, we derived and analyzed an equation linking the biases $B_{\mathrm{MW}}$, $B_{I}$, and $B_{C}\;$that was similar to the expression used in the framework of the Fast-PPP approach. It was proven that the slant ionosphere in this equation is no longer lumped with the DCB term $K_{21}$ and biased by the considerably smaller in magnitude Differential Phase Bias. This rather attractive property opens up an opportunity to directly retrieve accurate slant ionosphere estimates.

We then demonstrated some results illustrating the usefulness of the proposed parameterization for static user positioning and ionosphere information retrieval. Additionally, we analyzed the stability of the satellite-specific part of the carrier phase bias ${ϐ}_{\mathrm{MWC}}$. This bias is the complete analog of the wide-lane bias $b_{W}$ and can be estimated in exactly the same way as $b_{W}$. The importance of the stability analysis stems from the fact that the satellite-specific bias ${ϐ}_{\mathrm{MWC}}^{s}$ is to be provided as a part of the correction information.

We demonstrated some results of static user positioning using data from 60 globally distributed IGS stations. For each station, we divided the daily measurements into eight 3 h-long time spans and, therefore, analyzed 480 position time series. The positioning performance was analyzed in terms of the positioning accuracy, the convergence, and the ambiguity resolution performance. It was demonstrated that the absolute errors were typically below 1–2 cm horizontal and 5–10 cm vertical and nearly instant convergence was achieved depending on the strategy to handle the bias ${ϐ}_{\mathrm{MWC}}^{s}$, with the number of correctly fixed ambiguities exceeding 90%. Analysis of the daily and sub-daily (1 h- and 3 h-long) time series of the bias ${ϐ}_{\mathrm{MWC}}^{s}$ revealed its long- and short-term stability for both GPS and GALILEO. Furthermore, it was shown that a 3-h time interval may be sufficient for the bias update. This problem needs to be analyzed more carefully, though. A massive investigation of the impact of the bias update interval on the positioning performance is required. We also demonstrated a good potential for using the ionosphere retrieval Equation (22) associated with the proposed parameterization for ionosphere estimation. The comparison with different GIM TEC estimates showed that an agreement of 1–2 TECU with the standard deviation of 3–4 TECU can be reached.

It is worth noticing here that the results presented only serve to demonstrate the proposed clock parameterization concept. Correspondingly, they can be considered preliminary ones. There are many topics that need to be investigated to improve the understanding of the proposed idea and better identify its strong and weak aspects. First of all, a rigorous and detailed comparative analysis of the afore-described model and establishment of its connection to the existing PPP-RTK methods are required. Other important topics are the kinematic positioning performance with the afore-described user positioning model, and the extension of this model to GLONASS transmitting FDMA signals. Furthermore, the in-depth investigation of the performance of the slant ionosphere retrieval with Equation (22) linking the biases $B_{\mathrm{MW}}$, $B_{I},$ and $B_{C}$ and the analysis of the possibility to use this equation for convergence acceleration will be necessary. These problems will be the subject of future works.

At the same time, now we can identify and briefly outline the advantages and disadvantages the proposed parameterization delivers. For instance, it is advantageous that (1) the necessity to estimate and employ just one carrier phase bias can be in equal measure used in different PPP and PPP-RTK models based on ionosphere-free, single-, and multi-frequency undifferenced carrier phase measurements and, at the same time, easily estimated using the Melbourne-Wübenna combination; (2) the equality of the wide-lane and ionosphere-free biases leads to a new opportunity to estimate slant ionosphere with higher accuracy compared with the Fast-PPP; (3) there is a possibility to develop dual-frequency PPP-RTK models not requiring code hardware biases as in models with the conventional clock parameterization. At the same time, the most essential disadvantage is the necessity to provide the user the satellite DCBs in multi-frequency positioning models and, moreover, force the user to cope with their receiver counterparts as well. In the meantime, the proposed clock parameterization is not a unique one. It is, of course, possible to come up with other parameterizations giving rise to PPP and PPP-RTK models having other interesting and useful properties.

## Figures and Tables

**Figure 1 sensors-22-03117-f001:**
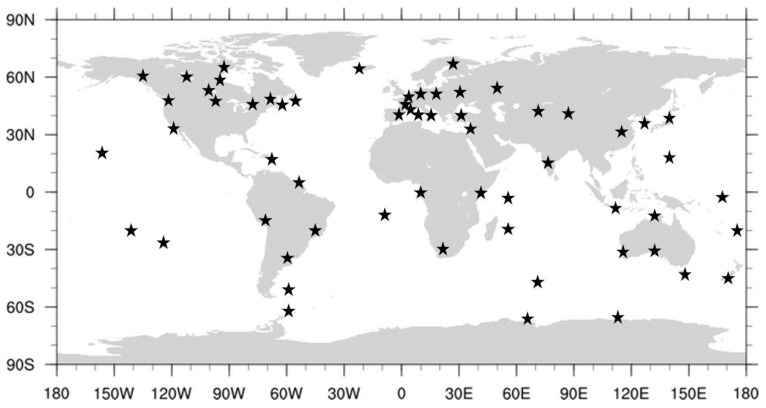
Distribution of the 60 IGS stations, the data of which were used in this study to generate the corrections.

**Figure 2 sensors-22-03117-f002:**
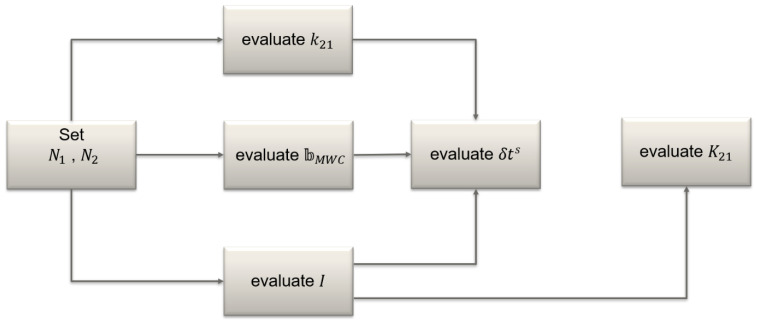
Flowchart of experimental correction generation.

**Figure 3 sensors-22-03117-f003:**
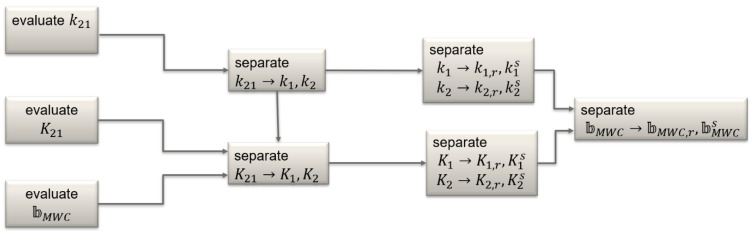
Flowchart of bias correction generation.

**Figure 4 sensors-22-03117-f004:**
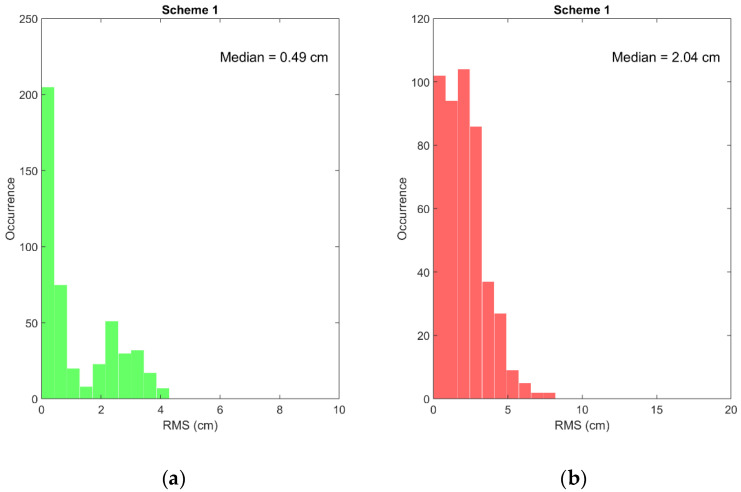
Distribution of the RMS positioning errors obtained with the proposed dual-frequency PPP-RTK model for the horizontal (**a**,**c**,**e**) and vertical (**b**,**d**,**f**) components for the three different processing schemes: Scheme 1 (**a**,**b**), Scheme 2 (**c**,**d**), and Scheme 3 (**e**,**f**).

**Figure 5 sensors-22-03117-f005:**
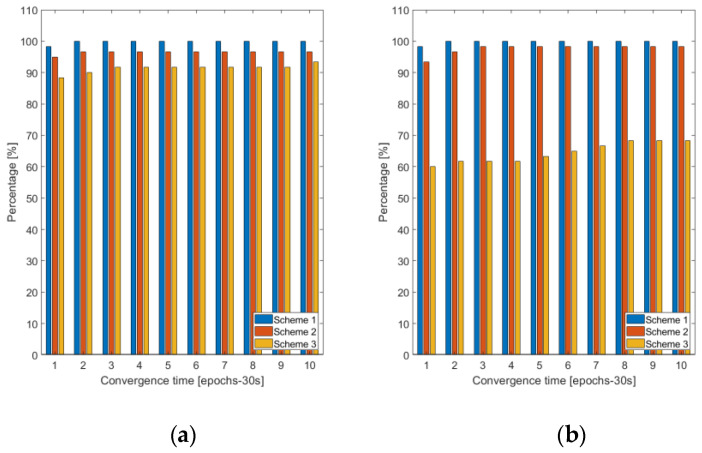
Cumulative histogram illustrating the convergence of the GNSS data to attain 10 cm horizontal (**a**) and vertical (**b**) for the three processing schemes.

**Figure 6 sensors-22-03117-f006:**
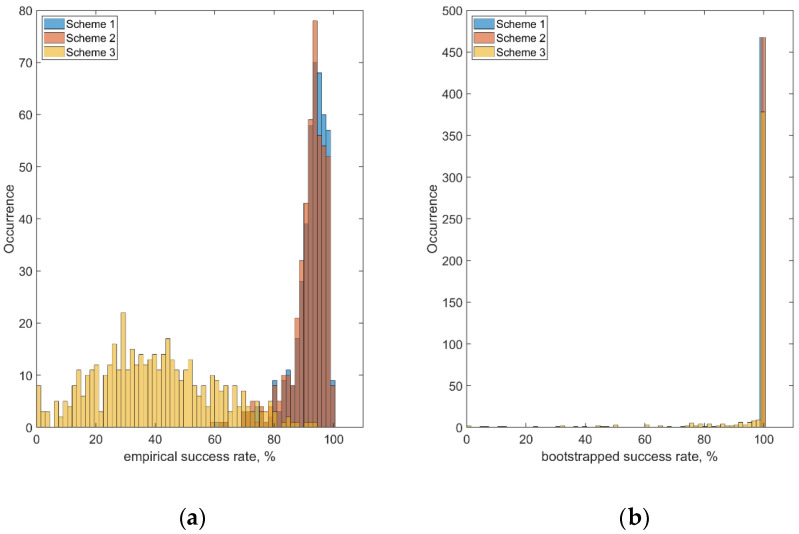
Distribution of the ambiguity resolution success rates for the three empirical (**a**) and bootstrapped (**b**) processing schemes.

**Figure 7 sensors-22-03117-f007:**
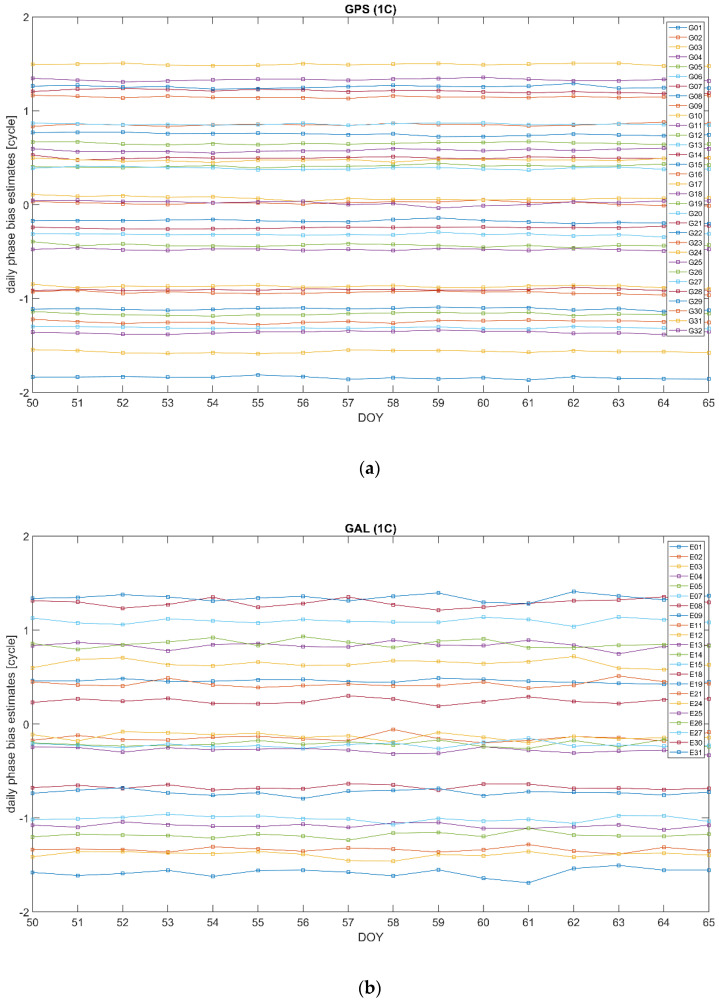
Daily bias ${ϐ}_{\mathrm{MWC}}^{s}$ estimates over a 16−day time interval 2020/50–2020/65 for GPS L1 (**a**) and GALILEO E1 (**b**). The time series for individual satellites are shifted along the y-axis by PRN−16 times 0.1. “1C” corresponds to L1 C/A for GPS and E1 pilot channel for GALILEO.

**Figure 8 sensors-22-03117-f008:**
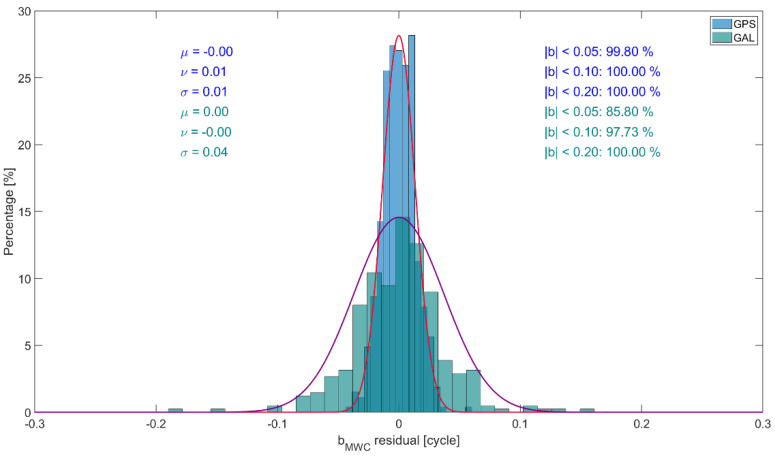
Distribution of the residuals of the daily ${ϐ}_{\mathrm{MWC}}^{s}$ estimates for GPS and GALILEO w.r.t. the overall 16−day mean. Red and purple curves show the approximating normal distributions for GPS and GALILEO, respectively.

**Figure 9 sensors-22-03117-f009:**
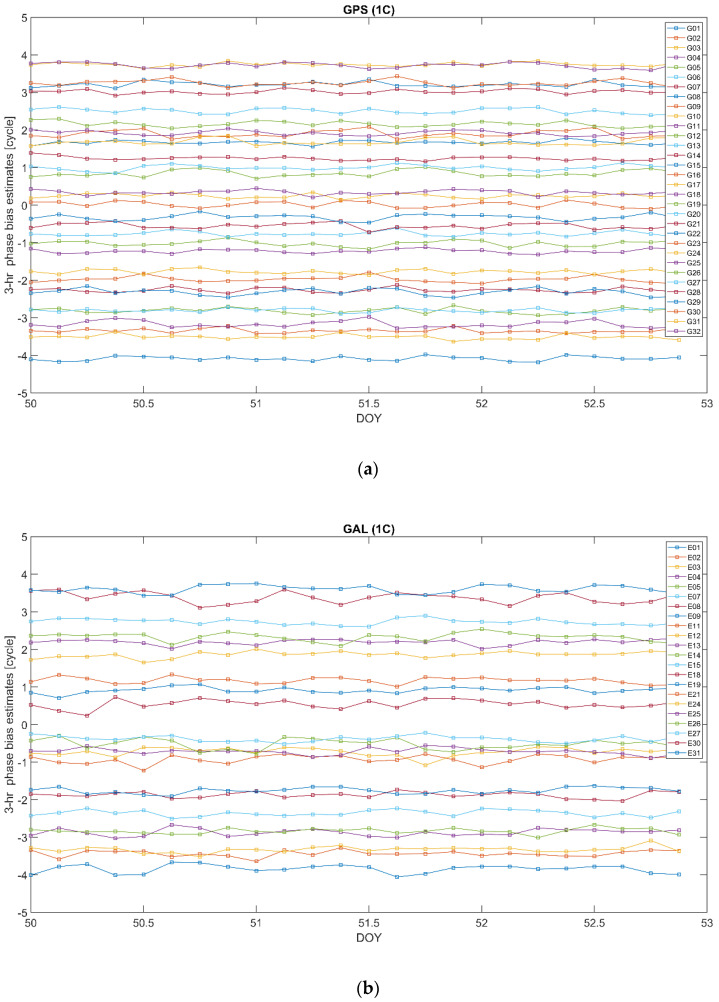
Sub−daily (3 h−long) bias ${ϐ}_{\mathrm{MWC}}^{s}$ estimates over a 3−day time interval 2020/50–2020/52 for GPS L1 (**a**) and GALILEO E1 (**b**). The time series for individual satellites are shifted along the y−axis by PRN−16 times 0.25. “1C” corresponds to L1 C/A for GPS and E1 pilot channel for GALILEO.

**Figure 10 sensors-22-03117-f010:**
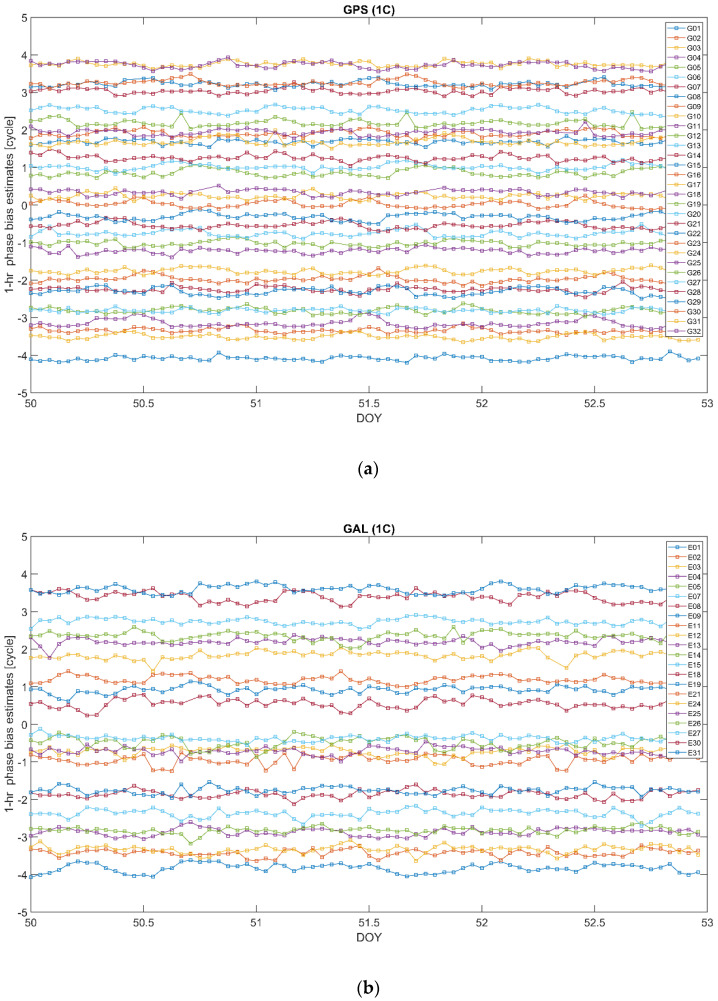
Sub−daily (hourly) bias ${ϐ}_{\mathrm{MWC}}^{s}$ estimates over a 3−day time interval 2020/50–2020/52 for GPS L1 (**a**) and GALILEO E1 (**b**). The time series for individual satellites are shifted along the y-axis by PRN−16 times 0.25. “1C” corresponds to L1 C/A for GPS and E1 pilot channel for GALILEO.

**Figure 11 sensors-22-03117-f011:**
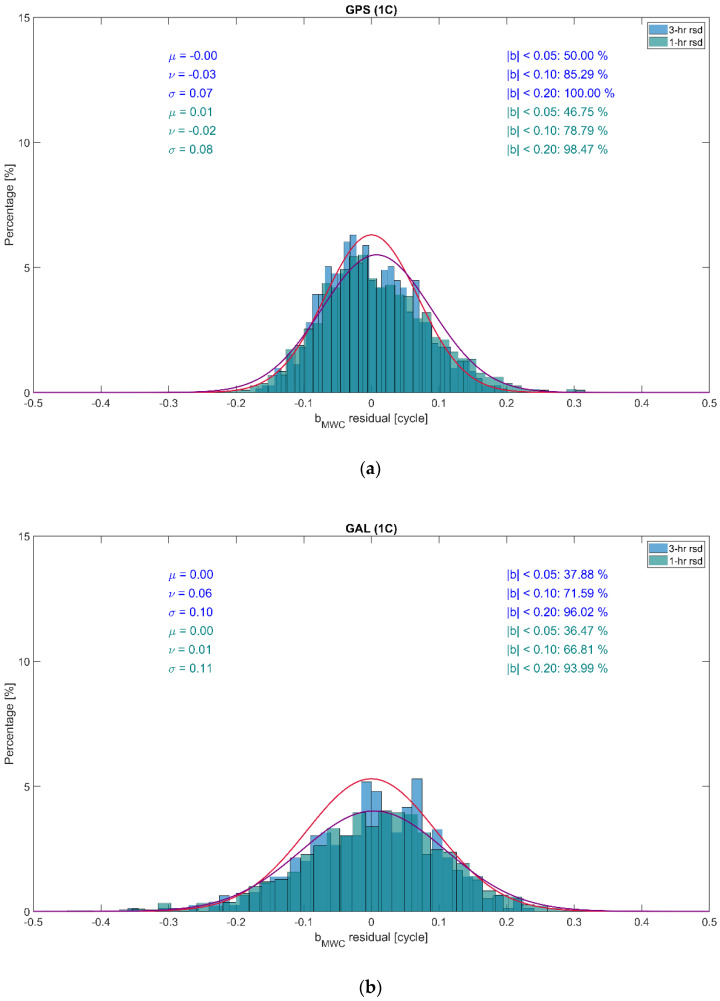
Distributions of the residuals of the sub-daily ${ϐ}_{\mathrm{MWC}}^{s}$ estimates for GPS (**a**) and GALILEO (**b**). Red and purple curves show the approximating normal distributions of 3−h and 1−h long sub-daily bias estimates, respectively.

**Figure 12 sensors-22-03117-f012:**
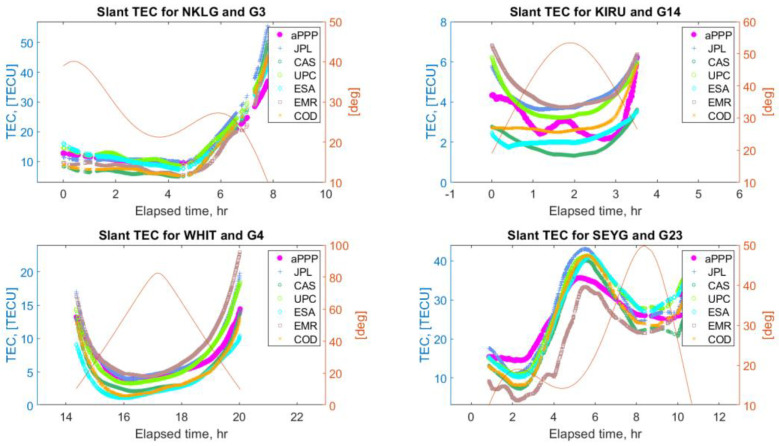
Comparison of STEC values retrieved using the ionosphere retrieval equation (labeled as aPPP) and GIM STEC values from six different Ionosphere Associate Analysis Centers (IAAC) for IGS stations NKLG, KIRU, WHIT, SEYG, and GPS satellites G3, G14, G4, and G23. The elevation angles of the corresponding satellites are also plotted.

**Figure 13 sensors-22-03117-f013:**
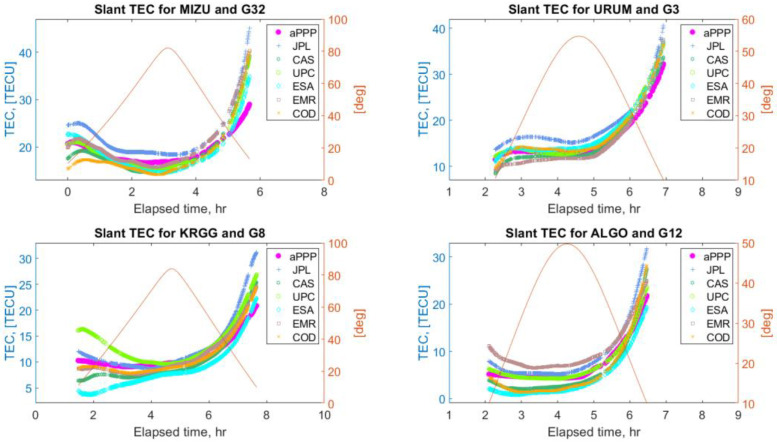
Comparison of the STEC values retrieved using the ionosphere retrieval equation (labeled as aPPP) and GIM STEC values from six different Ionosphere Associate Analysis Centers (IAAC) for IGS stations MIZU, URUM, KRGG, ALGO, and GPS satellites G32, G3, G8, and G12. Elevation angles of the corresponding satellites are also plotted.

**Figure 14 sensors-22-03117-f014:**
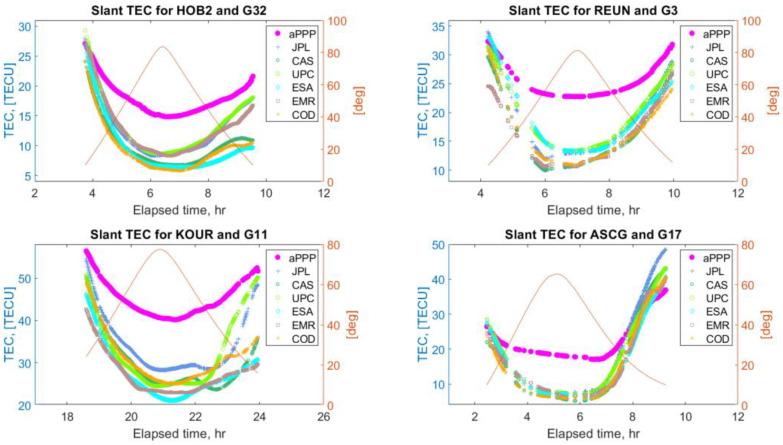
Comparison of STEC values retrieved using the ionosphere retrieval equation (labeled as aPPP) and GIM STEC values from six different Ionosphere Associate Analysis Centers (IAAC) for IGS stations HOB2, REUN, KOUR, ASCG, and GPS satellites G32, G3, G11, and G17. The elevation angles of the corresponding satellites are also plotted.

**Figure 15 sensors-22-03117-f015:**
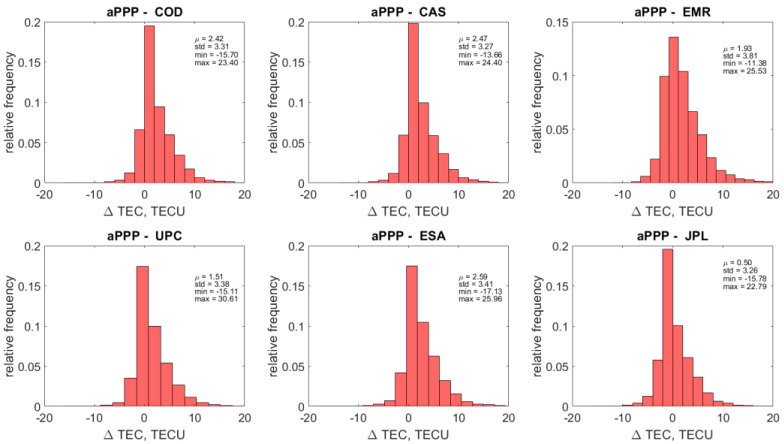
Distribution of the differences between Vertical TEC values retrieved using the ionosphere retrieval equation (labeled as aPPP) and GIM VTEC values from six different Ionosphere Associate Analysis Centers (IAAC) for GPS.

**Figure 16 sensors-22-03117-f016:**
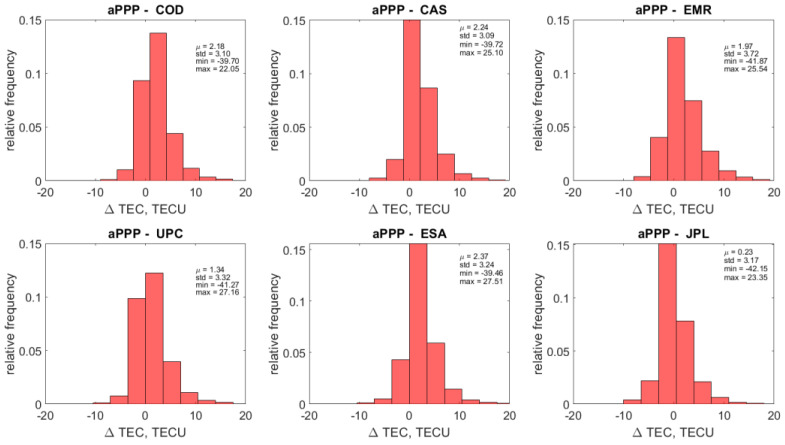
Distribution of the differences between Vertical TEC values retrieved using the ionosphere retrieval equation (labeled as aPPP) and GIM VTEC values from six different Ionosphere Associate Analysis Centers (IAAC) for GALILEO.

**Table 1 sensors-22-03117-t001:** A summary of the carrier phase biases in PPP models associated with the conventional clock parameterization (1).

Bias	Conventional Parameterization
L1 carrier phase bias	$b_{1\;}{=k}_{1}-{\;K}_{1}+2{\tilde{\alpha }}_{1}K_{21},\;{\tilde{\alpha }}_{1}=\frac{f_{2}^{2}}{f_{1}^{2}-f_{2}^{2}}$
L2 carrier phase bias	$b_{2}{=k}_{2}-{\;K}_{2\;}+{2\tilde{\alpha }}_{2}K_{21},\;{\tilde{\alpha }}_{2\;}=\frac{f_{1}^{2}}{f_{1\;}^{2}-{\;f}_{2}^{2}}$
Wide-lane bias	$b_{W}=\frac{f_{1}b_{1\;}-{\;f}_{2}b_{2}}{f_{1}-{\;f}_{2}}$
Ionosphere-free bias	$b_{C}=\frac{f_{1}^{2}b_{1}-{\;f}_{2}^{2}b_{2}}{f_{1\;}^{2}-{\;f}_{2}^{2}}$
Geometry-free bias	$b_{I}{=b}_{1\;}-{\;b}_{2}=\frac{1}{\alpha_{w}}\left(b_{W\;}-{\;b}_{C}\right)$

**Table 2 sensors-22-03117-t002:** Comparison of the carrier phase biases for the two clock parameterizations (1) and (14).

Bias	Conventional Parameterization	Proposed Parameterization
L1 bias	$b_{1}{=k}_{1}-K_{1}{+2\tilde{\alpha }}_{1}K_{21}$	${ϐ}_{1}={\tilde{\alpha }}_{1}k_{21}{+ϐ}_{\mathrm{MWC}},\;{\tilde{\alpha }}_{1}=\frac{f_{2}^{2}}{f_{1}^{2}-f_{2}^{2}}$
L2 bias	$b_{2}{=k}_{2}-K_{2}+2{\tilde{\alpha }}_{2}K_{21}$	${ϐ}_{2}={\tilde{\alpha }}_{2}k_{21}{+ϐ}_{\mathrm{MWC}},\;{\tilde{\alpha }}_{2}=\frac{f_{1}^{2}}{f_{1}^{2}-f_{2}^{2}}$
Wide-lane bias	$b_{W}$	${ϐ}_{W}=-\frac{f_{1}}{f_{1}{+f}_{2}}K_{1}-\frac{f_{2}}{f_{1}{+f}_{2}}K_{2}+\frac{f_{1}}{f_{1}-f_{2}}k_{1}-\frac{f_{2}}{f_{1}-f_{2}}k_{2}-\alpha_{w}k_{21}$ ${ϐ}_{W}{=b}_{W}-\alpha_{w}k_{21}{=ϐ}_{\mathrm{MWC}}-\alpha_{w}k_{21}$
Ionosphere-free bias	$b_{C}$	${ϐ}_{C}{=ϐ}_{\mathrm{MWC}}=-\frac{f_{1}}{f_{1}{+f}_{2}}K_{1}-\frac{f_{2}}{f_{1}{+f}_{2}}K_{2}+\frac{f_{1}}{f_{1}-f_{2}}k_{1}-\frac{f_{2}}{f_{1}-f_{2}}k_{2}$ ${ϐ}_{c}{=b}_{C}-\alpha_{w}k_{21}-\alpha_{w}K_{21}$
Melbourne-Wübenna bias	$b_{\mathrm{MW}}{=b}_{W}$	${ϐ}_{\mathrm{MW}}{=b}_{W}{=ϐ}_{\mathrm{MWC}}=-\frac{f_{1}}{f_{1}{+f}_{2}}K_{1}-\frac{f_{2}}{f_{1}{+f}_{2}}K_{2}+\frac{f_{1}}{f_{1}-f_{2}}k_{1}-\frac{f_{2}}{f_{1}-f_{2}}k_{2}$ ${ϐ}_{W}{=ϐ}_{\mathrm{MWC}}-\alpha_{w}k_{21}\;⟹{ϐ}_{\mathrm{MW}}\neq {ϐ}_{W}\;!$
Geometry-free bias	$b_{W}-b_{C}{=\alpha }_{w}b_{I}$	${ϐ}_{\mathrm{MW}}-{ϐ}_{C}{=\alpha }_{w}{ϐ}_{I}\equiv 0⟹{ϐ}_{I}\equiv 0\;!$

**Table 3 sensors-22-03117-t003:** Data processing strategy, observation models, and estimated parameters for network processing and user positioning.

Item	Models
Constellations	GPS+GALILEO
Procedure	Corrections generation: network processing/user positioning: PPP-RTK
Estimator	User positioning: KF
Observations	Undifferenced carrier phases, narrow-lane code
Signal selection	GPS: L1/L2; GAL: E1/E5a
Sampling interval	$30^{s}$
cutoff angle	10°
Phase wind-up	Applied [17]
Tropospheric delay	UNB3 m [18] with GMF [19]
Ionospheric delay	Evaluated using GIM as a-priori information, provided as corrections
Receiver clock	Estimated on the user side, random walk
Station displacement	Solid Earth tide, polar tide, ocean loading tide: IERS Conventions 1996 [20]
Terrestrial frame	ITRF 2014 [21]
Satellite orbit	CODE final orbits
Satellite clock	CODE 30-sec clocks, adjusted to the new parameterization and provided as corrections
Phase ambiguities	Network processing: set to integers/user positioning: LAMBDA PAR
Station coordinates	Network processing: precise in ITRF14/user positioning: estimated
Satellite PCO	Corrected using IGS values

**Table 4 sensors-22-03117-t004:** Global ionosphere maps used in the analysis.

GIM ID	Ionosphere Maps from
CAS	Chinese Academy of Science
CODE	Center for Orbit Determination in Europe, Bern, Switzerland
EMR	National Resources Canada
ESA	European Space Agency
JPL	Jet Propulsion Laboratory, Pasadena, USA
UPC	Polytechnic University of Catalonia, Barcelona

## Data Availability

The GNSS observation data were obtained from the online archives of NASA Crustal Dynamics Data Information System (CDDIS), https://cddis.nasa.gov/archive/gnss/data (accessed on 16 March 2022). The orbit and clock products were obtained from the online archives of the Center for Orbit Determination in Europe (CODE) of the Astronomical Institute at the University of Bern, ftp://ftp.aiub.unibe.ch/CODE_MGEX/CODE.

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
