# Peer review of "A Novel Clock Parameterization and Its Implications for Precise Point Positioning and Ionosphere Estimation"

_sensors, 2022, doi:10.3390/s22093117_

Round 1

Reviewer 1 Report

The whole article freely flowing style of writing, from the new function model to the advanced nature of the new model one by one. The relevant experimental results are favorable to support the author's conclusion, and I suggest employment. But at the same time, I raise the following small questions:

1.“Our TEC estimates demonstrated a promising agreement at a level of 1-2 TECU and the standard deviation of 3-4 TECU with GIM TEC values.“I have a question. The difference between the STEC value shown in Figure 14 and the GIM product can reach more than 10 TECUs. Why is it so large? So many products are basically the same, but the difference in this paper is so large, does it indicate that the proposed method has some defects? Is your conclusion a little hasty.

  1. The text clarity of Figure 1 is not enough. Would you consider adjusting it.

Reviewer 2 Report

It is an interesting study to provide a novel perspective on the clock parameterization. I hope the following comments can help improve the manuscript.

  1. The theoretical parts of the paper can be improved to make the main idea more explicit. Since there are so many equations and symbols, I recommend the authors to make sure that all the equation are correctly expressed and the meaning of each symbol is definite and clear. For example, the different meanings of B, b and the curlicue letter b should be explained carefully to avoid the reader’s misunderstanding.
  2. In Figure 4, the labels of the vertical axes should be “occurrence” (like Figure 6) or something, since the total numbers represented by all the histograms are 480 (am I right?).
  3. In Figure 11, the red curves have not been discussed or mentioned, and it seems that the two curves in each panel should be in different colors.
  4. For many times, the authors strengthen that the current study only give some primary results, and much work remains for the future. Therefore, it would be more valuable if some of the expected researches are implemented. I am looking forward to seeing a more comprehensive discussion.
  5. There are some typos or grammatical errors, please check all through the paper and polish the language. For example,

(1) Line 13: “by” should be “By”

(2) Line 77: demonstration -> demonstrating

(3) Line 566: The sentence “VTEC values…” seems incomplete

(4) Line 582: The phrase “As seen” should be “As is seen,”, or can be just deleted

(5) Line 591: “Also plotted are the elevation angles of corresponding satellites” will be better

Reviewer 3 Report

This paper is somehow complicated and serious. Why not to submit to gps solution or navigation or some similar journals that have more gnss readers? My major concern is why we can assume that b_MW and b_C are equal in Line 101. Authors did not justify this assumption in the paper. In my opinion, there's no reason why b_MW and b_C have to be equal. The validation can also be improved. The result comparisons are all internal, i.e., comparing with themselves. I would like to recommend the authors to compare with a regular ppp or a regular rtk without using this novel algorithm to justify the advantages. BTW, usually frequency is usually denoted by f instead of v, right?

Round 2

Reviewer 3 Report

I have no more questions.